# Logarithmic Regret for Online Control

**Naman Agarwal**[1]     **Elad Hazan**[1 2]     **Karan Singh**[1 2]
[1] Google AI Princeton
[2] Computer Science, Princeton University
namanagarwal@google.com, {ehazan,karans}@princeton.edu

## Abstract

We study optimal regret bounds for control in linear dynamical systems under adversarially changing strongly convex cost functions, given the knowledge of transition dynamics. This includes several well studied and fundamental frameworks such as the Kalman filter and the linear quadratic regulator. State of the art methods achieve regret which scales as $O(\sqrt{T})$, where $T$ is the time horizon.

We show that the optimal regret in this setting can be significantly smaller, scaling as $O(\text{poly}(\log T))$. This regret bound is achieved by two different efficient iterative methods, online gradient descent and online natural gradient.

## 1   Introduction

Algorithms for regret minimization typically attain one of two performance guarantees. For general convex losses, regret scales as square root of the number of iterations, and this is tight. However, if the loss function exhibit more curvature, such as quadratic loss functions, there exist algorithms that attain poly-logarithmic regret. This distinction is also known as "fast rates" in statistical estimation.

Despite their ubiquitous use in online learning and statistical estimation, logarithmic regret algorithms are almost non-existent in control of dynamical systems. This can be attributed to fundamental challenges in computing the optimal controller in the presence of noise.

Time-varying cost functions in dynamical systems can be used to model unpredictable dynamic resource constraints, and the tracking of a desired sequence of exogenous states. At a pinch, if we have changing (even, strongly) convex loss functions, the optimal controller for a linear dynamical system is not immediately computable via a convex program. For the special case of quadratic loss, some previous works [9] remedy the situation by taking a semi-definite relaxation, and thereby obtain a controller which has provable guarantees on regret and computational requirements. However, this semi-definite relaxation reduces the problem to regret minimization over linear costs, and removes the curvature which is necessary to obtain logarithmic regret.

In this paper we give the first efficient poly-logarithmic regret algorithms for controlling a linear dynamical system with noise in the dynamics (i.e. the standard model). Our results apply to general convex loss functions that are strongly convex, and not only to quadratics.

| Reference | Noise | Regret | loss functions |
|:---:|:---:|:---:|:---:|
| [1] | **none** | $O(\log^2 T)$ | quadratic (fixed hessian) |
| [4] | adversarial | $O(\sqrt{T})$ | convex |
| [9] | stochastic | $O(\sqrt{T})$ | quadratic |
| **here** | stochastic | $O(\log^7 T)$ | strongly convex |

## 1.1 Our Results

The setting we consider is a linear dynamical system, a continuous state Markov decision process with linear transitions, described by the following equation:

$$x_{t+1} = Ax_t + Bu_t + w_t. \tag{1.1}$$

Here $x_t$ is the state of the system, $u_t$ is the action (or control) taken by the controller, and $w_t$ is the noise. In each round $t$, the learner outputs an action $u_t$ upon observing the state $x_t$ and incurs a cost of $c_t(x_t, u_t)$, where $c_t$ is convex. The objective here is to choose a sequence of adaptive controls $u_t$ so that a minimum total cost may be incurred.

The approach taken by [9] and other previous works is to use a semi-definite relaxation for the controller. However, this removes the properties associated with the curvature of the loss functions, by reducing the problem to an instance of online linear optimization. It is known that without curvature, $O(\sqrt{T})$ regret bounds are tight (see [13]).

Therefore we take a different approach, initiated by [4]. We consider controllers that depend on the previous noise terms, and take the form $u_t = \sum_{i=1}^{H} M_i w_{t-i}$. While this resulting convex relaxation does not remove the curvature of the loss functions altogether, it results in an overparametrized representation of the controller, and it is not a priori clear that the loss functions are strongly convex with respect to the parameterization. We demonstrate the appropriate conditions on the linear dynamical system under which the strong convexity is retained.

Henceforth we present two methods that attain poly-logarithmic regret. They differ in terms of the regret bounds they afford and the computational cost of their execution. The online gradient descent update (OGD) requires only gradient computation and update, whereas the online natural gradient (ONG) update, in addition, requires the computation of the preconditioner, which is the expected Gram matrix of the Jacobian, denoted $J$, and its inverse. However, the natural gradient update admits an instance-dependent upper bound on the regret, which while being at least as good as the regret bound on OGD, offers better guarantees on benign instances (See Corollary 4.5, for example).

| Algorithm | Update rule (simplified) | Applicability |
|-----------|--------------------------|---------------|
| OGD | $M_{t+1} \leftarrow M_t - \eta_t \nabla f_t(M_t)$ | $\exists K$, diag $L$ s.t. $A - BK = QLQ^{-1}$ |
| ONG | $M_{t+1} \leftarrow M_t - \eta_t (\mathbb{E}[J^\top J])^{-1} \nabla f_t(M_t)$ | $\|L\| \leq 1 - \delta, \|Q\|, \|Q\|^{-1} \leq \kappa$ |

## 1.2 Related Work

For a survey of linear dynamical systems (LDS), as well as learning, prediction and control problems, see [17]. Recently, there has been a renewed interest in learning dynamical systems in the machine learning literature. For fully-observable systems, sample complexity and regret bounds for control (under Gaussian noise) were obtained in [3, 10, 2]. The technique of spectral filtering for learning and open-loop control of partially observable systems was introduced and studied in [15, 7, 14]. Provable control in the Gaussian noise setting via the policy gradient method was also studied in [11].

The closest work to ours is that of [1] and [9], aimed at controlling LDS with adversarial loss functions. The authors in [3] obtain a $O(\log^2 T)$ regret algorithm for changing quadratic costs (with a fixed hessian), but for dynamical systems that are noise-free. In contrast, our results apply to the full (noisy) LDS setting, which presents the main challenges as discussed before. Cohen et al. [9] consider changing quadratic costs with stochastic noise to achieve a $O(\sqrt{T})$ regret bound.

We make extensive use of techniques from online learning [8, 16, 13]. Of particular interest to our study is the setting of online learning with memory [5]. We also build upon the recent control work of [4], who use online learning techniques and convex relaxation to obtain provable bounds for LDS with adversarial perturbations.

## 2 Problem Setting

We consider a linear dynamical system as defined in (1.1) with costs $c_t(x_t, u_t)$, where $c_t$ is strongly convex. In this paper we assume that the noise $w_t$ is a random variable generated independently at

every time step. For any algorithm $\mathcal{A}$, we attribute a cost defined as

$$J_T(\mathcal{A}) = \mathbb{E}_{\{w_t\}} \left[ \sum_{t=1}^{T} c_t(x_t, u_t) \right],$$

where $x_{t+1} = Ax_t + Bu_t + w_t$, $u_t = \mathcal{A}(x_1, \ldots x_t)$ and $\mathbb{E}_{\{w_t\}}$ represents the expectation over the entire noise sequence. For the rest of the paper we will drop the subscript $\{w_t\}$ from the expectation as it will be the only source of randomness. Overloading notation, we shall use $J_T(K)$ to denote the cost of a linear controller $K$ which chooses the action as $u_t = -Kx_t$.

**Assumptions.** In the paper we assume that $x_1 = 0$ [1], as well as the following conditions.

**Assumption 2.1.** *We assume that $\|B\| \leq \kappa_B$. Furthermore, the perturbation introduced per time step is bounded, i.i.d, and zero-mean with a lower bounded covariance i.e.*

$$\forall t \; w_t \sim \mathcal{D}_w, \mathbb{E}[w_t] = 0, \mathbb{E}[w_t w_t^\top] \succeq \sigma^2 I \text{ and } \|w_t\| \leq W$$

This may be adapted to the case of sub-gaussian noise by conditioning on the event that none of the noise vectors are ever large. Such adaptation introduces a multiplicative $\log(T)$ factor in the regret.

**Assumption 2.2.** *The costs $c_t(x, u)$ are $\alpha$-strongly convex. Wehnever $\|x\|, \|u\| \leq D$, it holds that*

$$\|\nabla_x c_t(x, u)\|, \|\nabla_u c_t(x, u)\| \leq GD.$$

The class of linear controllers we work with are defined as follows; see Section A for a detailed note.

**Definition 2.3** (Diagonal Strong Stability). *Given a dynamics $(A, B)$, a linear controller $K$ is $(\kappa, \gamma)$-diagonal strongly stable for real numbers $\kappa \geq 1, \gamma < 1$, if there exists a complex diagonal matrix $L$ and a non-singular complex matrix $Q$, such that $A - BK = QLQ^{-1}$ with the following being true:*

1. *The spectral norm of $L$ is strictly smaller than one, i.e., $\|L\| \leq 1 - \gamma$.*

2. *The controller and transforming matrices are bounded, i.e., $\|K\| \leq \kappa$ and $\|Q\|, \|Q^{-1}\| \leq \kappa$.*

**Regret Formulation.** Let $\mathcal{K} = \{K : K \text{ is } (\kappa, \gamma)\text{-diagonal strongly stable}\}$. For an algorithm $\mathcal{A}$, the notion of regret we consider is *pseudo-regret*, i.e. the sub-optimality of its cost with respect to the cost for the best linear controller i.e.,

$$\texttt{Regret} = J_T(\mathcal{A}) - \min_{K \in \mathcal{K}} J_T(K).$$

## 3 Preliminaries

**Notation.** We reserve the letters $x, y$ for states and $u, v$ for actions. We denote by $d_x, d_u$ to be the dimensionality of the state and the control space respectively. Let $d = \max(d_x, d_u)$. We reserve capital letters $A, B, K, M$ for matrices associated with the system and the policy. Other capital letters are reserved for universal constants in the paper. We use the shorthand $M_{i:j}$ to denote a subsequence $\{M_i, \ldots, M_j\}$. For any matrix $U$, define $U_{\text{vec}}$ to be a flattening of the matrix where we stack the columns upon each other. Further for a collection of matrices $M = \{M^{[i]}\}$, let $M_{vec}$ be the flattening defined by stacking the flattenings of $M^{[i]}$ upon each other. We use $\|x\|_U^2 = x^\top U x$ to denote the matrix induced norm. The rest of this section provides a recap of the relevant definitions and concepts introduced in [4].

### 3.1 Reference Policy Class

For the rest of the paper, we fix a $(\kappa, \gamma)$-diagonally strongly stable matrix $\mathbb{K}$ (The bold notation is to stress that we treat this matrix as fixed and not a parameter). Note that this can be any such matrix and it can be computed via a semi-definite feasibility program [9] given the knowledge of the dynamics, before the start of the game. We work with following the class of policies.

**Definition 3.1** (Disturbance-Action Policy). *A disturbance-action policy $M = (M^{[0]}, \ldots, M^{[H-1]})$, for horizon $H \geq 1$ is defined as the policy which at every time $t$, chooses the recommended action $u_t$ at a state $x_t$, defined [2] as*

$$u_t(M) \triangleq -\mathbb{K}x_t + \sum_{i=1}^{H} M^{[i-1]} w_{t-i}.$$

*For notational convenience, here it may be considered that $w_i = 0$ for all $i < 0$.*

The policy applies a linear transformation to the disturbances observed in the past $H$ steps. Since $(x, u)$ is a linear function of the disturbances in the past under a linear controller $K$, formulating the policy this way can be seen as a relaxation of the class of linear policies. Note that $\mathbb{K}$ is a fixed matrix and is not part of the parameterization of the policy. As was established in [4] (and we include the proof for completeness), with the appropriate choice of parameters, superimposing such a $\mathbb{K}$, to the policy class allows it to approximate any linear policy in terms of the total cost suffered with a finite horizon parameter $H$.

We refer to the policy played at time $t$ as $M_t = \{M_t^{[i]}\}$ where the subscript $t$ refers to the time index and the superscript $[i-1]$ refers to the action of $M_t$ on $w_{t-i}$. Note that such a policy can be executed because $w_{t-1}$ is perfectly determined on the specification of $x_t$ as $w_{t-1} = x_t - Ax_{t-1} - Bu_{t-1}$.

## 3.2 Evolution of State

This section describes the evolution of the state of the linear dynamical system under a non-stationary policy composed of a sequence of $T$ policies, where at each time the policy is specified by $M_t = (M_t^{[0]}, \ldots, M_t^{[H-1]})$. We will use $M_{0:T-1}$ to denote such a non-stationary policy. The following definitions ease the burden of notation.

1. Define $\tilde{A} = A - B\mathbb{K}$. $\tilde{A}$ shall be helpful in describing the evolution of state starting from a non-zero state in the absence of disturbances.

2. For any sequence of matrices $M_{0:H}$, define $\Psi_i$ as a linear function that describes the effect of $w_{t-i}$ on the state $x_t$, formally defined below.

**Definition 3.2.** *For any sequence of matrices $M_{0:H}$, define the disturbance-state transfer matrix $\Psi_i$ for $i \in \{0, 1, \ldots H\}$, to be a function with $h + 1$ inputs defined as*

$$\Psi_i(M_{0:H}) \triangleq \tilde{A}^i \mathbf{1}_{i \leq H} + \sum_{j=0}^{H} \tilde{A}^j B M_{H-j}^{[i-j-1]} \mathbf{1}_{i-j \in [1,H]}.$$

It will be important to note that $\psi_i$ is a **linear** function of its argument.

## 3.3 Surrogate State and Surrogate Cost

This section introduces a couple of definitions required to describe our main algorithm. In essence they describe a notion of state, its derivative and the expected cost if the system evolved solely under the past $H$ steps of a non-stationary policy.

**Definition 3.3** (Surrogate State & Surrogate Action). *Given a sequence of matrices $M_{0:H+1}$ and $2H$ independent invocations of the random variable $w$ given by $\{w_j \sim \mathcal{D}_w\}_{j=0}^{2H-1}$, define the following random variables denoting the surrogate state and the surrogate action:*

$$y(M_{0:H}) = \sum_{i=0}^{2H} \Psi_i(M_{0:H}) w_{2H-i-i},$$

$$v(M_{0:H+1}) = -\mathbb{K}y(M_{0:H}) + \sum_{i=1}^{H} M_{H+1}^{[i-1]} w_{2H-i}.$$

*When $M$ is the same across all arguments we compress the notation to $y(M)$ and $v(M)$ respectively.*

**Algorithm 1** Online Control Algorithm

---
1: **Input:** Step size schedule $\eta_t$, Parameters $\kappa_B, \kappa, \gamma, T$.
2: Define $H = \gamma^{-1} \log(T\kappa^2)$
3: Define $\mathcal{M} = \{M = \{M^{[0]} \ldots M^{[H-1]}\} : \|M^{[i-1]}\| \leq \kappa^3 \kappa_B (1-\gamma)^i\}$.
4: Initialize $M_0 \in \mathcal{M}$ arbitrarily.
5: **for** $t = 0, \ldots, T-1$ **do**
6:      Choose the action:
$$u_t = -\mathbb{K}x_t + \sum_{i=1}^{H} M_t^{[i-1]} w_{t-i}.$$
7:      Observe the new state $x_{t+1}$ and record $w_t = x_{t+1} - Ax_t - Bu_t$.
8:      **Online Gradient Update:**
$$M_{t+1} = \Pi_{\mathcal{M}}(M_t - \eta_t \nabla f_t(M_t))$$

9:      **Online Natural Gradient Update:**
$$M_{vec,t+1} = \Pi_{\mathcal{M}}(M_{vec,t} - \eta_t (\mathbb{E}[J^T J])^{-1} \nabla_{M_{vec,t}} f_t(M_t))$$

10: **end for**

---

**Definition 3.4** (Surrogate Cost). *Define the surrogate cost function $f_t$ to be the cost associated with the surrogate state-action pair defined above, i.e.,$f_t(M_{0:H+1}) = \mathbb{E}\left[c_t(y(M_{0:H}), v(M_{0:H+1}))\right]$. When $M$ is the same across all arguments we compress the notation to $f_t(M)$.*

**Definition 3.5** (Jacobian). *Let $z(M) = \begin{bmatrix} y(M) \\ v(M) \end{bmatrix}$. Since $y(M), v(M)$ are random linear functions of $M$, $z(M)$ can be reparameterized as $z(M) = JM_{vec} = \begin{bmatrix} J_y \\ J_v \end{bmatrix} M_{vec}$, where $J$ is a random matrix, which derives its randomness from the random perturbations $w_i$.*

### 3.4 OCO with Memory

We now describe the setting of online convex optimization with memory introduced in [5]. In this setting, at every step $t$, an online player chooses some point $x_t \in \mathcal{K} \subset \mathbb{R}^d$, a loss function $f_t : \mathcal{K}^{H+1} \mapsto \mathbb{R}$ is then revealed, and the learner suffers a loss of $f_t(x_{t-H:t})$. We assume a certain coordinate-wise Lipschitz regularity on $f_t$ of the form such that, for any $j \in \{0, \ldots, H\}$, for any $x_{0:H}, \tilde{x}_j \in \mathcal{K}$,

$$|f_t(x_{0:j-1}, x_j, x_{j+1:H}) - f_t(x_{0:j-1}, \tilde{x}_j, x_{j+1:H})| \leq L\|x_j - \tilde{x}_j\|. \tag{3.1}$$

In addition, we define $f_t(x) = f_t(x, \ldots, x)$, and we let

$$G_f = \sup_{t \in \{0, \ldots, T\}, x \in \mathcal{K}} \|\nabla f_t(x)\|, \quad D = \sup_{x,y \in \mathcal{K}} \|x - y\|. \tag{3.2}$$

The resulting goal is to minimize the *policy regret* [6], which is defined as

$$\texttt{PolicyRegret} = \sum_{t=H}^{T} f_t(x_{t-H:t}) - \min_{x \in \mathcal{K}} \sum_{t=H}^{T} f_t(x).$$

## 4 Algorithms & Statement of Results

The two variants of our method are spelled out in Algorithm 1. Theorems 4.1 and 4.3 provide the main guarantees for the two algorithms.

**Online Gradient Update**

**Theorem 4.1** (Online Gradient Update). *Suppose Algorithm 1 (Online Gradient Update) is executed with $\mathbb{K}$ being any $(\kappa, \gamma)$-diagonal strongly stable matrix and $\eta_t = \Theta\left(\alpha\sigma^2 t\right)^{-1}$, on an LDS satisfying Assumption 2.1 with control costs satisfying Assumption 2.2. Then, it holds true that*

$$J_T(\mathcal{A}) - \min_{K \in \mathcal{K}} J_T(K) \leq \tilde{O}\left(\frac{G^2 W^4}{\alpha\sigma^2} \log^7(T)\right).$$

The above result leverages the following lemma which shows that the function $f_t(\cdot)$ is strongly convex with respect to its argument $M$. Note that strong convexity of the cost functions $c_t$ over the state-action space does not by itself imply the strong convexity of the surrogate cost $f_t$ over the space of controllers $M$. This is because, in the surrogate cost $f_t$, $c_t$ is applied to $y(M), v(M)$ which themselves are linear functions of $M$; the linear map $M$ is necessarily column-rank-deficient. To observe this, note that $M$ maps from a space of dimensionality $H \times \dim(x) \times \dim(u)$ to that of $\dim(x) + \dim(u)$. The next theorem, which forms the core of our analysis, shows that this is not the case using the inherent stochastic nature of the dynamical system.

**Lemma 4.2.** *If the cost functions $c_t(\cdot, \cdot)$ are $\alpha$-strongly convex, $\mathbb{K}$ is a $(\kappa, \gamma)$ diagonal strongly stable matrix and Assumption 2.1 is met then the idealized functions $f_t(M)$ are $\lambda$-strongly convex with respect to $M$ where*

$$\lambda = \frac{\alpha \sigma^2 \gamma^2}{36 \kappa^{10}}$$

We present the proof for simple cases in Section 6, deferring the general proof to Section F.

**Online Natural Gradient Update**

**Theorem 4.3** (Online Natural Gradient Update). *Suppose Algorithm 1 (Online Natural Gradient Update) is executed with $\eta_t = \Theta (\alpha t)^{-1}$, on an LDS satisfying Assumptions 2.1 and with control costs satisfying Assumption 2.2. Then, it holds true that*

$$J_T(\mathcal{A}) - \min_{K \in \mathcal{K}} J_T(K) \leq \tilde{O} \left( \frac{GW^2}{\alpha \mu} \log^7(T) \right) \quad where \quad \mu^{-1} \triangleq \max_{M \in \mathcal{M}} \|(\mathbb{E}[J^T J])^{-1} \nabla_{M_{vec}} f_t(M)\|.$$

In Theorem 4.3, the regret guarantee depends on an instance-dependent parameter $\mu$, which is a measure of hardness of the problem. First, we note that the proof of Lemma 4.2 establishes that the Gram matrix of the Jacobian (Defintion 3.5) is strictly positive definite and hence we recover the logarithmic regret guarantee achieved by the Online Gradient Descent Update, with the constants preserved.

**Corollary 4.4.** *In addition to the assumptions in Theorem 4.3, if $\mathbb{K}$ is a $(\kappa, \gamma)$-diagonal strongly stable matrix, then for the natural gradient update*

$$J_T(\mathcal{A}) - \min_{K \in \mathcal{K}} J_T(K) \leq \tilde{O} \left( \frac{G^2 W^4}{\alpha \sigma^2} \log^7(T) \right),$$

*Proof.* The conclusion follows from Lemma 5.2 and Lemma 6.1 which is the core component in the proof of Lemma 4.2 showing that $\mathbb{E}[J^T J] \geq \frac{\gamma^2 \sigma^2}{36 \kappa^{10}} \cdot \mathbb{I}$. $\square$

Secondly, we note that, being instance-dependent, the guarantee the Natural Gradient update offers can potentially be stronger than that of the Online Gradient method. A case in point is the following corollary involving spherically symmetric quadratic costs, in which case the Natural Gradient update yields a regret guarantee under demonstrably more general conditions, in that the bound does not depend on the minimum eigenvalue of the covariance of the disturbances $\sigma^2$, unlike OGD.

**Corollary 4.5.** *Under the assumptions on Theorem 4.3, if the cost functions are of the form $c_t(x, u) = r_t(\|x\|^2 + \|u\|^2)$, where $r_t \in [\alpha, \beta]$ is an adversarially chosen sequence of numbers and $\mathbb{K}$ is chosen to be a $(\kappa, \gamma)$-diagonal strongly stable matrix, then the natural gradient update guarantees*

$$J_T(\mathcal{A}) - \min_{K \in \mathcal{K}} J_T(K) \leq \tilde{O} \left( \frac{\beta^2 W^2}{\alpha} \log^7(T) \right),$$

*Proof.* Note $\|\nabla_{M_{vec}} f_t(M)\|_{(\mathbb{E}[J^T J])^{-2}} = \|\mathbb{E}[J^T (r_t \cdot I) J M_{vec}]\|_{(\mathbb{E}[J^T J])^{-2}} \leq \beta \|M_{vec}\|.$ $\square$

## 5 Regret Analysis

The next section is a condensation of the results from [4] which we present in this form to highlight the reduction to OCO with memory.

## 5.1 Reduction to Low Regret with Memory

The next lemma shows that achieving low policy regret on the memory based function $f_t$ is sufficient to ensure low regret on the overall dynamical system. Since the proof is essentially provided by [4], we provide it in the Appendix for completeness. Define,

$$\mathcal{M} \triangleq \{M = \{M^{[0]} \dots M^{[H-1]}\} : \|M^{[i-1]}\| \leq \kappa^3 \kappa_B (1-\gamma)^i\}.$$

**Lemma 5.1.** *Let the dynamical system satisfy Assumption 2.1 and let $\mathbb{K}$ be any $(\kappa, \gamma)$-diagonal strongly stable matrix. Consider a sequence of loss functions $c_t(x, u)$ satisfying Assumption 2.2 and a sequence of policies $M_0 \dots M_T$ satisfying*

$$\texttt{PolicyRegret} = \sum_{t=0}^{T} f_t(M_{t-H-1:t}) - \min_{M \in \mathcal{M}} \sum_{t=0}^{T} f_t(M) \leq R(T)$$

*for some function $R(T)$ and $f_t$ as defined in Definition 3.4. Let $A$ be an online algorithm that plays the non-stationary controller sequence $\{M_0, \dots M_T\}$. Then as long as $H$ is chosen to be larger than $\gamma^{-1} \log(T\kappa^2)$ we have that*

$$J(A) - \min_{K^* \in \mathcal{K}} J(K^*) \leq R(T) + O(GW^2 \log(T)),$$

*Here $O(\cdot)$, $\Theta(\cdot)$ contain polynomial factors in $\gamma^{-1}, \kappa_B, \kappa, d$.*

**Lemma 5.2.** *The function $f_t$ as defined in Definition 3.4 is coordinate-wise L-lipschitz and the norm of the gradient is bounded by $G_f$, where*

$$L = \frac{2DGW\kappa_B\kappa^3}{\gamma}, \quad G_f \leq GDWHd\left(H + \frac{2\kappa_B\kappa^3}{\gamma}\right)$$

$$\text{where } D \triangleq \frac{W\kappa^2(1 + H\kappa_B^2\kappa^3)}{\gamma(1 - \kappa^2(1-\gamma)^{H+1})} + \frac{\kappa_B\kappa^3 W}{\gamma}.$$

The proof of this lemma is identical to the analogous lemma in [4] and hence is omitted.

## 5.2 Analysis for Online Gradient Descent

In the setting of Online Convex Optimization with Memory, as shown by [5], by running a memory-based OGD, we can bound the policy regret by the following theorem, proven in the appendix.

**Theorem 5.3.** *Consider the OCO with memory setting defined in Section 3.4. Let $\{f_t\}_{t=H}^{T}$ be Lipschitz loss functions with memory such that $f_t(x)$ are $\lambda$-strongly convex, and let $L$ and $G_f$ be as defined in (3.1) and (3.2). Then, there exists an algorithm which generates a sequence $\{x_t\}_{t=0}^{T}$ such*

$$\sum_{t=H}^{T} f_t(x_{t-H:t}) - \min_{x \in \mathcal{K}} \sum_{t=H}^{T} \tilde{f}_t(x) \leq \frac{G_f^2 + LH^2 G_f}{\lambda}(1 + \log(T)).$$

*Proof of Theorem 4.1.* Setting $H = \gamma^{-1}\log(T\kappa^2)$, Theorem 5.3, in conjunction with Lemma 5.2, implies that policy regret is bounded by $\tilde{O}\left(\frac{G^2 W^4 H^6}{\alpha\sigma^2}\log T\right)$. An invocation of Lemma 5.1 now suffices to conclude the proof of the claim. $\qquad\square$

## 5.3 Analysis for Online Natural Gradient Descent

Consider structured loss functions of the form $f_t(M_{0:H+1}) = \mathbb{E}[c_t(z)]$, where $z = \sum_{i=0}^{H+1} J_i[M_i]_{\text{vec}}$. $J_i$ is a random matrix, and $c_t$'s are adversarially chosen strongly convex loss functions. In a similar vein, define $f_t(M)$ to be the specialization of $f_t$ when input the same argument, i.e. $M$, $H+1$ times. Define $J = \sum_{i=0}^{H+1} J_i$. The proof of the following theorem may be found in the appendix.

**Theorem 5.4.** *In the setting desribed in this subsection, let $c_t$ be $\alpha$-strongly convex, and $f_T$ be such that it satisfies equation (3.1) with constant $L$, and $G_f = \max_{M \in \mathcal{M}} \|(\mathbb{E}[J^T J])^{-1} \nabla_{M_{vec}} f_t(M)\|$. Then, the online natural gradient update generates a sequence $\{M_t\}_{t=0}^{T}$ such that*

$$\sum_{t=H}^{T} f_t(M_{t-H:t}) - \min_{M \in \mathcal{M}} \sum_{t=H}^{T} \tilde{f}_t(M) \leq \frac{\max_{M \in \mathcal{M}} \|\nabla_{M_{vec}} f_t(M)\|_{(\mathbb{E}[J^T J])^{-1}}^2 + LH^2 G_f}{\alpha}(1 + \log(T)).$$

*Proof of Theorem 4.3.* First observe that $\|\nabla_{M_{\text{vec}}} f_t(M)\|^2_{(\mathbb{E}[J^T J])^{-1}} \leq \mu^{-1} \|\nabla_{M_{\text{vec}}} f_t(M)\|$. Setting $H = \gamma^{-1} \log(T\kappa^2)$, Theorem 5.4, in conjunction with Lemma 5.2, imply the stated bound on policy regret. An invocation of Lemma 5.1 suffices to conclude the proof of the claim. □

## 6   Proof of Strong Convexity in Simpler Cases

We will need some definitions and preliminaries that are outlined below. By definition we have that $f_t(M) = \mathbb{E}[c_t(y_t(M), v_t(M))]$. Since we know that $c_t$ is strongly convex we have that

$$\nabla^2 f_t(M) = \mathbb{E}_{\{w_k\}^{2H-1}_{k=0}}[\nabla^2 c_t(y(M), v(M))] \succeq \alpha \mathbb{E}_{\{w_k\}^{2H-1}_{k=0}}[J_y^\top J_y + J_v^\top J_v].$$

$J_y, J_v$ are random matrices dependent on the noise $\{w_k\}^{2H-1}_{k=0}$. The next lemma implies Lemma 4.2.

**Lemma 6.1.** *If Assumption 2.1 is satisfied and $\mathbb{K}$ is chosen to be a $(\kappa, \gamma)$-diagonal strongly stable matrix, then the following holds,*

$$\mathbb{E}_{\{w_k\}^{2H-1}_{k=0}}[J_y^\top J_y + J_v^\top J_v] \succeq \frac{\gamma^2 \sigma^2}{36\kappa^{10}} \cdot \mathbb{I}.$$

To analyze $J_y, J_v$, we will need to rearrange the definition of $y(M)$ to make the dependence on each individual $M^{[i]}$ explicit. To this end consider the following definition for all $k \in [H+1]$.

$$\tilde{v}_k(M) \triangleq \sum_{i=1}^{H} M^{[i-1]} w_{2H-i-k}$$

Under this definition it follows that

$$y(M) = \sum_{k=1}^{H} (A - B\mathbb{K})^{k-1} B \tilde{v}_k(M) + \sum_{k=1}^{H} (A - B\mathbb{K})^{k-1} w_{2H-k}$$

$$v(M) = -\mathbb{K}y(M) + \tilde{v}_0(M)$$

From the above definitions, $(J_y, J_v)$ may be characterized in terms of the Jacobian of $\tilde{v}_k$ with respect to $M$, which we define for the rest of the section as $J_{\tilde{v}_k}$. Defining $M_{\text{vec}}$ as the stacking of rows of each $M^{[i]}$ vertically, i.e. stacking the columns of $(M^{[i]})^\top$, it can be observed that for all $k$,

$$J_{\tilde{v}_k} = \frac{\partial \tilde{v}_k(M)}{\partial M} = \begin{bmatrix} I_{d_u} \otimes w_{2H-k-1}^\top & I_{d_u} \otimes w_{2H-k-2}^\top & \cdots & I_{d_u} \otimes w_{H-k}^\top \end{bmatrix}$$

where $d_u$ is the dimension of the controls. We are now ready to analyze the two simpler cases. Further on in the section we drop the subscripts $\{w_k\}^{2H-1}_{k=0}$ from the expectations for brevity.

### 6.1   Proof of Lemma 6.1: $\mathbb{K} = 0$

In this section we assume that $\mathbb{K} = 0$ is a $(\kappa, \gamma)$-diagonal strongly stable policy for $(A, B)$. Be definition, we have $v(M) = \tilde{v}_0(M)$. One may conclude the proof with the following observation.

$$\mathbb{E}[J_y^\top J_y + J_v^\top J_v] \succeq \mathbb{E}[J_v^\top J_v] = \mathbb{E}[J_{\tilde{v}_0}^\top J_{\tilde{v}_0}] = I_{d_u} \otimes \Sigma \succeq \sigma^2 \mathbb{I}.$$

### 6.2   Proof of Lemma 6.1: 1-dimensional case

Here, we specialize Lemma 4.2 to one-dimensional state and one-dimensional control. This case highlights the difficulty caused in the proof due to a choosing a non-zero $\mathbb{K}$ and presents the main ideas of the proof in a simplified notation.

Note that in the one dimensional case, the policy given by $M = \{M^{[i]}\}^{H-1}_{i=0}$ is an $H$ dimensional vector with $M^{[i]}$ being a scalar. Furthermore $y(M), v(M), \tilde{v}_k(M)$ are scalars and hence their Jacobians $J_y, J_v, J_{\tilde{v}_k}$ with respect to $M$ are $1 \times H$ vectors. In particular we have that,

$$J_{\tilde{v}_k} = \frac{\partial \tilde{v}_k(M)}{\partial M} = \begin{bmatrix} w_{2H-k-1} & w_{2H-k-2} & \cdots & w_{H-k} \end{bmatrix}$$

Therefore using the fact that $E[w_i w_j] = 0$ for $i \neq j$ and $\mathbb{E}[w_i^2] = \sigma^2$, it can be observed that for any $k_1, k_2$, we have that

$$\mathbb{E}[J_{v_{\tilde{k}_1}}^\top J_{v_{\tilde{k}_2}}] = \mathcal{T}_{k_1 - k_2} \cdot \sigma^2 \tag{6.1}$$

where $\mathcal{T}_m$ is defined as an $H \times H$ matrix with $[\mathcal{T}_m]_{ij} = 1$ if and only if $i - j = m$ and 0 otherwise. This in particular immediately gives us that,

$$\mathbb{E}[J_y^\top J_y] = \underbrace{\left( \sum_{k_1=1}^{H} \sum_{k_2=1}^{H} \mathcal{T}_{k_1 - k_2} \cdot (A - B\mathbb{K})^{k_1 - 1 + k_2 - 1} \right)}_{\triangleq \mathbb{G}} \cdot B^2 \cdot \sigma^2 \tag{6.2}$$

$$\mathbb{E}[J_{\tilde{v}_0}^\top J_y] = \underbrace{\left( \sum_{k=1}^{H} \mathcal{T}_{-k} (A - B\mathbb{K})^{k-1} \right)}_{\triangleq \mathbb{Y}} \cdot B \cdot \sigma^2 \tag{6.3}$$

First, we prove a few spectral properties of the matrices $\mathbb{G}$ and $\mathbb{Y}$ defined above. From Gershgorin's circle theorem, and the fact that $\mathbb{K}$ is $(\kappa, \gamma)$-diagonal strongly stable, we have

$$\|\mathbb{Y} + \mathbb{Y}^\top\| \leq \| \sum_{k=1}^{H} (\mathcal{T}_{-k} + \mathcal{T}_k)(A - B\mathbb{K})^{k-1} \| \leq 2\gamma^{-1} \tag{6.4}$$

The spectral properties of $\mathbb{G}$ summarized in the lemma below form the core of our analysis.

**Lemma 6.2.** $\mathbb{G}$ *is a symmetric positive definite matrix. In particular* $\mathbb{G} \succeq \frac{1}{4} \cdot I$.

Now consider the statements which follow by the respective definitions.

$$\mathbb{E}[J_v^\top J_v] = \mathbb{K}^2 \cdot \mathbb{E}[J_y^\top J_y] - \mathbb{K} \cdot \mathbb{E}[J_y^\top J_{\tilde{v}_0}] - \mathbb{K} \cdot \mathbb{E}[J_{\tilde{v}_0}^\top J_y] + \mathbb{E}[J_{\tilde{v}_0}^\top J_{\tilde{v}_0}]$$
$$= \sigma^2 \cdot \underbrace{\left( B^2 \mathbb{K}^2 \cdot \mathbb{G} - B\mathbb{K} \cdot (\mathbb{Y} + \mathbb{Y}^\top) + I \right)}_{\triangleq \mathbb{F}}.$$

Now $\mathbb{F} \succeq 0$. We finish the proof by considering two cases. The first case is when $3|B|\gamma^{-1}\kappa \geq 1$. Noting $\kappa \geq 1$, in this case Lemma 6.2 immediately implies that

$$m^\top \left( \mathbb{F} + B^2 \cdot \mathbb{G} \right) m \geq m^\top \left( B^2 \cdot \mathbb{G} \right) m \geq \frac{\frac{1}{4}\|m\|^2}{9\gamma^{-2}\kappa^2} \geq \frac{\gamma^2\|m\|^2}{36\kappa^{10}},$$

In the second case (when $3|B|\gamma^{-1}\kappa \leq 1$), (6.4) implies that

$$m^\top \left( \mathbb{F} + B^2 \cdot \mathbb{G} \right) m \geq m^\top \left( I - B\mathbb{K} \cdot (\mathbb{Y} + \mathbb{Y}^\top) \right) m \geq (1/3)\|m\|^2 \geq \frac{\gamma^2\|m\|^2}{36\kappa^{10}}.$$

### 6.2.1 Proof of Lemma 6.2

Recall $\mathcal{T}_m$ is defined as an $H \times H$ matrix with $[\mathcal{T}_m]_{ij} = 1$ if and only if $i - j = m$ and 0 otherwise. Define the following matrix for any complex number $|\psi| < 1$.

$$\mathbb{G}(\psi) = \sum_{k_1=1}^{H} \sum_{k_2=1}^{H} \mathcal{T}_{k_1 - k_2} \left( \psi^\dagger \right)^{k_1 - 1} \psi^{k_2 - 1}$$

Note that $\mathbb{G}$ in Lemma 6.2 is equal to $\mathbb{G}(A - B\mathbb{K})$. The following lemma, proven in Section E, provides a lower bound on the spectrum of the matrix $\mathbb{G}(\psi)$. The lemma presents the proof of a more general case ($\phi$ is complex) that aids the multi-dimensional case. A special case when $\phi = 1$ was proven in [12], and we follow a similar approach relying on the inverse.

**Lemma 6.3.** *Let $\psi$ be a complex number such that $|\psi| \leq 1$. We have that $\mathbb{G}(\psi) \succeq (1/4) \cdot I_H$.*

## 7 Conclusion

We presented two algorithms for controlling linear dynamical systems with strongly convex costs with regret that scales poly-logarithmically with time. This improves state-of-the-art known regret bounds that scale as $O(\sqrt{T})$. It remains open to extend the poly-log regret guarantees to more general systems and loss functions, such as exp-concave losses, or alternatively, show that this is impossible.

**Acknowledgements**

The authors thank Sham Kakade and Cyril Zhang for various thoughtful discussions. Elad Hazan acknowledges funding from NSF grant # CCF-1704860.

## Footnotes

[1]This is only for convenience of presentation. The case with a bounded $x_1$ can be handled similarly.

[2] $x_t$ is completely determined given $w_0 \ldots w_{t-1}$. Hence, the use of $x_t$ only serves to ease the burden of presentation.

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
