[Supplementary Material]

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

# Appendix

## A    Discussion on Diagonal Strong Stability

Classically, a controller $K$ is stabilizing [17] if the spectral radius of $A - BK \leq 1 - \delta$. The notion of strong stability was introduced by [9] – being the same as Definition 2.3, but not requiring $L$ to be diagonal. Both strong stability and diagonal strong stability are quantitative measures of the classical notion of stabilizing controllers that permit a discussion on non-asymptotic regret bounds. We note that an analogous notion for quantification of open-loop stability appears in the work of [14].

On the generality of the diagonal strong stability notion, the following comment may be made: while not all matrices are complex diagonalizable, an exhaustive characterization of $m \times m$ complex diagonal matrices is the existence of $m$ linearly independent eigenvectors; for the later, it suffices, but is not necessary, that a matrix has $m$ distinct eigenvalues (See [18]). It may be observed that almost all matrices admit distinct eigenvalues, and hence, are complex diagonalizable insofar the complement set admits a zero-measure. By this discussion, *almost all* stabilizing controllers are diagonal strongly stable for some $\kappa, \gamma$. The astute reader may note the departure here from the more general notion – strongly stability – in that *all* stabilizing controllers are strongly stable for some choice of parameters.

## B    Proof of the Reductions to OCO with Memory

Since the proof of Lemma will borrow heavily from the definitions introduced by [4], we restate those definitions here for convenience. Please note that some of these definitions overload our previous definitions but it will be clear from the context.

### B.1    Definitions

1. Let $x_t^K(M_{0:t-1})$ is the state attained by the system upon execution of a non-stationary policy $\pi(M_{0:t-1}, K)$. We similarly define $u_t^K(M_{0:t-1})$ to be the action executed at time $t$. If the same policy $M$ is used across all time steps, we compress the notation to $x_t^K(M), u_t^K(M)$. Note that $x_t^K(0), u_t^K(0)$ refers to running the linear policy $K$.

2. $\Psi_{t,i}^{K,h}(M_{t-h:t})$ is a transfer matrix that describes the effect of $w_{t-i}$ with respect to the past $h + 1$ policies on the state $x_{t+1}$, formally defined below. When $M$ is the same across all arguments we compress the notation to $\Psi_{t,i}^{K,h}(M)$.

**Definition B.1.** *For any $t, h \leq t, i \leq H + h$, define the disturbance-state transfer matrix $\Psi_{t,i}^{K,h}$ to be a function with $h + 1$ inputs defined as*

$$\Psi_{t,i}^{K,h}(M_{t-h:t}) = \tilde{A}_K^i \mathbf{1}_{i \leq h} + \sum_{j=0}^{h} \tilde{A}_K^j B M_{t-j}^{[i-j-1]} \mathbf{1}_{i-j \in [1,H]}.$$

**Definition B.2** (Surrogate State & Surrogate Action). *Define,*

$$y_{t+1}^K(M_{t-H:t}) = \sum_{i=0}^{2H} \Psi_{t,i}^{K,H}(M_{t-H:t}) w_{t-i},$$

$$v_{t+1}^K(M_{t-H:t+1}) = -K y_{t+1}^K(M_{t-H:t}) + \sum_{i=1}^{H} M_{t+1}^{[i-1]} w_{t+1-i}.$$

*When $M$ is the same across all arguments we compress the notation to $y_{t+1}^K(M), v_{t+1}^K(M)$.*

**Definition B.3** (Surrogate Cost). *Define the surrogate cost function $f_t$ to be the cost associated with the surrogate state and surrogate action, i.e.,*

$$f_t(M_{t-H-1:t}) = \mathbb{E}\left[ c_t(y_t^K(M_{t-H-1:t-1}), v_t^K(M_{t-H-1:t})) \right].$$

*When $M$ is the same across all arguments we compress the notation to $f_t(M)$.*

Note that this definition coincides exactly with Definition 3.4 in the main text.

## B.2 Prerequisites

In this section we state some lemmas and theorems which were proved in [4]. Due to consistency of definitions the proofs of these are omitted and can be found in [4].

**Lemma B.4** (Sufficiency). *For any two $(\kappa, \gamma)$-diagonal strongly stable matrices $K^*, K$, there exists $M_* = (M_*^{[0]}, \ldots, M_*^{[H-1]}) \in \mathcal{M}$ defined as*

$$M_*^{[i]} = (K - K^*)(A - BK^*)^i$$

*such that*

$$\sum_{t=0}^{T} \left( c_t(x_t^K(M_*), u_t^K(M_*)) - c_t(x_t^{K^*}(0), u_t^{K^*}(0)) \right) \leq T \cdot \frac{2GDWH\kappa_B^2\kappa^5(1-\gamma)^H}{\gamma}.$$

**Theorem B.5.** *For any $(\kappa, \gamma)$-diagonal strongly stable $K$, any $\tau > 0$, and any sequence of policies $M_1 \ldots M_T$ satisfying $\|M_t^{[i]}\| \leq \tau(1-\gamma)^i$, if the perturbations are bounded by $W$, we have that*

$$\sum_{t=1}^{T} f_t(M_{t-H-1:t}) - \sum_{t=1}^{T} c_t(x_t^K(M_{0:t-1}), u_t^K(M_{0:t})) \leq 2TGD^2\kappa^3(1-\gamma)^{H+1},$$

*where*

$$D \triangleq \frac{W\kappa^3(1 + H\kappa_B\tau)}{\gamma(1 - \kappa^2(1-\gamma)^{H+1})} + \frac{\tau W}{\gamma}.$$

## B.3 Proof of Lemma 5.1

*Proof of Lemma 5.1.* Let $D$ be defined as

$$D \triangleq \frac{W\kappa^3(1 + H\kappa_B\tau)}{\gamma(1 - \kappa^2(1-\gamma)^{H+1})} + \frac{\kappa_B\kappa^3 W}{\gamma}.$$

Let $K^*$ be the optimal linear policy in hindsight. By definition $K^*$ is a $(\kappa, \gamma)$-diagonal strongly stable matrix. Using Lemma B.4 and Theorem B.5, we have that

$$\min_{M_* \in \mathcal{M}} \left( \sum_{t=0}^{T} f_t(M_*) \right) - \sum_{t=0}^{T} c_t(x_t^{K^*}(0), u_t^{K^*}(0))$$

$$\leq \min_{M_* \in \mathcal{M}} \left( \sum_{t=0}^{T} c_t(x_t^K(M_*), u_t^K(M_*)) \right) - \sum_{t=0}^{T} c_t(x_t^{K^*}(0), u_t^{K^*}(0)) + 2TGD^2\kappa^3(1-\gamma)^{H+1}$$

$$\leq 2TGD(1-\gamma)^{H+1} \left( \frac{WH\kappa_B^2\kappa^5}{\gamma} + D\kappa^3 \right). \tag{B.1}$$

Note that by definition of $\mathcal{M}$, we have that

$$\forall t \in [T], \forall i \in [H] \quad \|M_t^{[i]}\| \leq \kappa_B\kappa^3(1-\gamma)^i.$$

Using Theorem B.5 we have that

$$\sum_{t=0}^{T} c_t(x_t^K(M_{0:t-1}), u_t^K(M_{0:t-1})) - \sum_{t=0}^{T} f_t(M_{t-H-1:t}) \leq 2TGD^2\kappa^3(1-\gamma)^{H+1}. \tag{B.2}$$

Summing up (B.1) and (B.2) and using the condition that $H \geq \frac{1}{\gamma}\log(T\kappa^2)$, we get the result. $\quad\square$

**Algorithm 2** OGD with Memory (OGD-M).

---
1: **Input:** Step size $\eta$, functions $\{f_t\}_{t=m}^T$
2: Initialize $x_0, \ldots, x_{H-1} \in \mathcal{K}$ arbitrarily.
3: **for** $t = H, \ldots, T$ **do**
4:    Play $x_t$, suffer loss $f_t(x_{t-H}, \ldots, x_t)$
5:    Set $x_{t+1} = \Pi_{\mathcal{K}} \left( x_t - \eta \nabla \tilde{f}_t(x) \right)$
6: **end for**

---

## C  Policy Regret for Online Gradient Descent

*Proof of Theorem 5.3.* By the standard OGD strong convexity analysis, if $\eta_t = (\lambda \cdot (t - H))^{-1}$, we have that

$$\sum_{t=H}^T \tilde{f}_t(x_t) - \min_{x \in \mathcal{K}} \sum_{t=H}^T \tilde{f}_t(x) \leq \frac{G^2}{2\lambda}(1 + \log(T)).$$

In addition, we know by (3.1) that, for any $t \geq H$,

$$|f_t(x_{t-H}, \ldots, x_t) - f_t(x_t, \ldots, x_t)| \leq L \sum_{j=1}^H \|x_t - x_{t-j}\| \leq L \sum_{j=1}^H \sum_{l=1}^j \|x_{t-l+1} - x_{t-l}\|$$

$$\leq L \sum_{j=1}^H \sum_{l=1}^j \eta_{t-l} \|\nabla \tilde{f}_{t-l}(x_{t-l})\| \leq LH^2 \eta_{t-H} G,$$

and so we have that

$$\left| \sum_{t=H}^T f_t(x_{t-H}, \ldots, x_t) - \sum_{t=H}^T f_t(x_t, \ldots, x_t) \right| \leq \frac{LH^2 G}{\lambda}(1 + \log(T)).$$

It follows that

$$\sum_{t=H}^T f_t(x_{t-H}, \ldots, x_t) - \min_{x \in \mathcal{K}} \sum_{t=H}^T f_t(x, \ldots, x) \leq \frac{G^2 + LH^2 G}{\lambda}(1 + \log(T)).$$

$\square$

## D  Policy Regret for Online Natural Gradient Descent

Recall the setting involving structured loss functions of the form $f_t(M_{0:H+1}) = \mathbb{E}[c_t(z)]$, where $z = \sum_{i=0}^{H+1} J_i [M_i]_{\text{vec}}$. $J_i$ is a random matrix, and $c_t$'s are adversarially chosen strongly convex loss functions. In a similar vein, define $f_t(M)$ to be the specialization of $f_t$ when input the same argument, i.e. $M$, $H + 1$ times. Define $J = \sum_{i=0}^{H+1} J_i$. The following lemma provides upper bounds on the regret bound as well as the norm of the movement of iterate at every round for the Online Natural Gradient Update (Algorithm 1).

**Lemma D.1.** *For $\alpha$-strongly convex $c_t$, if the iterates $M_t$ are chosen as per the update rule:*

$$[M_{t+1}]_{vec} = \Pi_{\mathcal{M}} \left( [M_t]_{vec} - \eta_t (\mathbb{E}[J^T J])^{-1} \nabla_{[M_t]_{vec}} f_t(M_t) \right)$$

*with a decreasing step size of $n_t = \frac{1}{\alpha t}$, it holds that*

$$\sum_{t=1}^T f_t(M_t) - \min_{M^* \in \mathcal{M}} \sum_{t=1}^T f_t(M^*) \leq (2\alpha)^{-1} \max_{M \in \mathcal{M}} \|\nabla_{M_{vec}} f_t(M)\|_{(\mathbb{E}[J^T J])^{-1}}^2 \log T.$$

*Moreover, the norm of the movement of consecutive iterates is bounded for all $t$ as*

$$\|[M_{t+1}]_{vec} - [M_t]_{vec}\| \leq (\alpha t)^{-1} \max_{M \in \mathcal{M}} \|(\mathbb{E}[J^T J])^{-1} \nabla_{M_{vec}} f_t(M)\|.$$

*Proof.* Let $M^* = \arg\min_{M\in\mathcal{M}} \sum_{t=1}^{T} f_t(M)$, $z_t = JM_{vec,t}$ and $z^* = JM_{\text{vec}}^*$. Now, we have, as consequence of strong convexity of $c_t$, that

$$\sum_{t=1}^{T} f_t(M_t) - \sum_{t=1}^{T} f_t(M^*) \leq \mathbb{E}\left[\langle \nabla_z c_t(z_t), z_t - z^*\rangle - \frac{\alpha}{2}\|z_t - z^*\|^2\right].$$

With $P = \mathbb{E}[J^T J]$, the choice of the update rule ensures that

$$\|[M_{t+1}]_{\text{vec}} - M_{\text{vec}}^*\|_P^2$$
$$= \|[M_t]_{\text{vec}} - M_{\text{vec}}^*\|_P^2 - 2\eta_t \langle \nabla_{[M_t]_{\text{vec}}} f_t(M_t), [M_t]_{\text{vec}} - M_{\text{vec}}^*\rangle + \eta_t^2 \|\nabla_{[M_t]_{\text{vec}}} f_t(M_t)\|_{P^{-1}}.$$

Observe by the application of chain rule and linearity of expectation that

$$\mathbb{E}[\langle \nabla_z c_t(z_t), z_t - z^*\rangle] = \mathbb{E}[\langle \nabla_z c_t(z_t), J([M_t]_{\text{vec}} - M_{\text{vec}}^*)\rangle]$$
$$= \langle \nabla_{[M_t]_{\text{vec}}} f_t(M_t), [M_t]_{\text{vec}} - M_{\text{vec}}^*\rangle,$$
$$\mathbb{E}[\|z_t - z^*\|^2] = \|[M_t]_{\text{vec}} - M_{\text{vec}}^*\|_P^2.$$

Combining these (in)equalities, we have

$$\sum_{t=1}^{T} f_t(M_t) - \sum_{t=1}^{T} f_t(M^*)$$
$$\leq \sum_{t=1}^{T}\left(\frac{\|[M_t]_{\text{vec}} - M_{vec}^*\|_P^2 - \|[M_{t+1}]_{\text{vec}} - M_{vec}^*\|_P^2}{2\eta_t} + \frac{\eta_t}{2}\|\nabla_{[M_t]_{\text{vec}}} f_t(M_t)\|_{P^{-1}}^2\right)$$
$$- \frac{\alpha}{2}\|[M_t]_{vec} - M_{vec}^*\|_P^2$$
$$\leq (2\alpha)^{-1} \max_{M\in\mathcal{M}} \|\nabla_{M_{\text{vec}}} f_t(M)\|_{P^{-1}}^2 \log T$$

$\square$

*Proof of Theorem 5.4.* We know by (3.1) that, for any $t \geq H$,

$$|f_t(M_{t-H:t}) - f_t(M)| \leq L\sum_{j=1}^{H}\|[M_t]_{\text{vec}} - [M_{t-j}]_{\text{vec}}\| \leq L\sum_{j=1}^{H}\sum_{l=1}^{j}\|[M_{t-l+1}]_{\text{vec}} - [M_{t-l}]_{\text{vec}}\|$$
$$\leq L\sum_{j=1}^{H}\sum_{l=1}^{j}\eta_{t-l}\max_{M\in\mathcal{M}}\|(\mathbb{E}[J^T J])^{-1}\nabla_{M_{\text{vec}}} f_t(M)\|$$
$$\leq LH^2\eta_{t-H}\max_{M\in\mathcal{M}}\|(\mathbb{E}[J^T J])^{-1}\nabla_{M_{\text{vec}}} f_t(M)\|,$$

and so we have that

$$\left|\sum_{t=H}^{T} f_t(M_{t-H:t}) - \sum_{t=H}^{T} f_t(M_t)\right| \leq \frac{LH^2 G_f}{\alpha}(1 + \log(T)).$$

The result follows by invoking Lemma D.1. $\square$

# E Spectral Lower Bound on $\mathbb{G}$

*Proof of Lemma 6.3.* The following definitions help us express the matrix $\mathbb{G}$ in a more convenient form. For any number $\psi \in \mathbb{C}$, such that $|\psi| < 1$ and any $h$ define,

$$S_\psi(h) = \sum_{i=1}^{h} |\psi|^{2(i-1)} = \frac{1 - |\psi|^{2h}}{1 - |\psi|^2}.$$

With the above definition it can be seen that the entries $\mathbb{G}(\psi)$ can be expressed in the following manner,

$$[\mathbb{G}(\psi)]_{ij} = S_\psi(H - |i - j|) \cdot \psi^{i-j} \qquad \text{if } j \geq i$$

$$[\mathbb{G}(\psi)]_{ij} = (\psi^\dagger)^{j-i} \cdot S_\psi(H - |i - j|) \qquad \text{if } i \geq j$$

Schematically the matrix $\mathbb{G}(\psi)$ looks like

$$\begin{bmatrix}
S_\psi(H) & S_\psi(H-1)\psi & S_\psi(H-2)\psi^2 & . & . & S(2)\psi^{H-2} & S(1)\psi^{H-1} \\
\psi^\dagger S_\psi(H-1) & S_\psi(H) & S_\psi(H-1)\psi & . & . & S(3)\psi^{H-3} & S(2)\psi^{H-2} \\
(\psi^\dagger)^2 S_\psi(H-2) & \psi^\dagger S_\psi(H-1) & S_\psi(H) & . & . & S(4)\psi^{H-4} & S(3)\psi^{H-3} \\
. & . & . & . & . & . & . \\
. & . & . & . & . & . & . \\
(\psi^\dagger)^{H-1} S_\psi(1) & (\psi^\dagger)^{H-2} S_\psi(2) & (\psi^\dagger)^{H-3} S_\psi(3) & . & . & \psi^\dagger S_\psi(H-1) & S_\psi(H)
\end{bmatrix}.$$

We analytically compute the inverse of the matrix $\mathbb{G}(\psi)$ below and bound its spectral norm.

**Claim E.1.** *The inverse of $\mathbb{G}(\psi)$ has the following form.*

$$[\mathbb{G}(\psi)]^{-1} = \begin{bmatrix}
\alpha & b & 0 & . & . & 0 & 0 & \beta^\dagger \\
b^\dagger & a & b & . & . & 0 & 0 & 0 \\
0 & b^\dagger & a & . & . & 0 & 0 & . \\
. & 0 & b^\dagger & . & . & b & 0 & . \\
. & 0 & 0 & . & . & a & b & 0 \\
0 & 0 & 0 & . & . & b^\dagger & a & b \\
\beta & 0 & 0 & . & . & 0 & b^\dagger & \alpha
\end{bmatrix},$$

*where the relevant quantities above are given by the following formula*

$$b = \frac{-\psi}{1 + |\psi|^{2H}} \qquad a = -b(\psi^\dagger + \psi^{-1}) = \frac{1 + |\psi|^2}{1 + |\psi|^{2H}}$$

$$\beta = \frac{(1 - |\psi|^2)}{(1 - (|\psi|^2)^{H+1})} \frac{(\psi^\dagger)^H \psi}{(1 + |\psi|^{2H})} \qquad \alpha = \frac{1 - (|\psi|^2)^{H+2}}{(1 - (|\psi|^2)^{H+1})(1 + (|\psi|^{2H}))}.$$

Since $|\psi| < 1$, it is easy to see that $|\alpha|, |a| \leq 2$ and $|\beta|, |b| \leq 1$. This immediately implies that $\|(\mathbb{G}(\psi))^{-1}\| \leq 4$ and therefore the lemma follows.

To prove the remnant claim, the following may be verified, implying $\mathbb{G}(\psi)[\mathbb{G}(\psi)]^{-1} = I$.

**Case A**: Let's first consider the diagonal entries and in particular $i = j \in [1, H-2]$. We have that

$$\left[\mathbb{G}(\psi)[\mathbb{G}(\psi)]^{-1}\right]_{i,i} = b \cdot \psi^\dagger S_\psi(H-1) + b^\dagger \cdot \psi S_\psi(H-1) + a S_\psi(H)$$

$$= \frac{-2|\psi|^2 S_\psi(H-1) + (1 + |\psi|^2) S_\psi(H)}{1 + |\psi|^{2H}} = 1$$

**Case B**: Lets consider the diagonal entry $(0, 0)$. (The $(H, H)$ entry is the complement and hence equal to 1).

$$\left[\mathbb{G}(\psi)[\mathbb{G}(\psi)]^{-1}\right]_{0,0}$$

$$= \alpha \cdot S_\psi(H) + b^\dagger \psi S_\psi(H-1) + \beta^\dagger (\psi^\dagger)^{H-1} S_\psi(1)$$

$$= \frac{(1 - (|\psi|^2)^{H+2}) S_\psi(H) - (1 - (|\psi|^2)^{H+1})|\psi|^2 S_\psi(H-1) + (1 - |\psi|^2)(|\psi|^{2H})}{(1 - (|\psi|^2)^{H+1})(1 + (|\psi|^{2H}))}$$

$$= 1$$

**Case C**: Now lets consider non diagonal entries, in particular for $j \in [1, H-2]$ and $i \in [0, H-1]$ and $i > j$. (The case with the same conditions and $j > i$ follows by replacing $\psi$ with $\psi^\dagger$ in the computation below)

$$\left[\mathbb{G}(\psi)[\mathbb{G}(\psi)]^{-1}\right]_{i,j} = (\psi^\dagger)^{i-j-1}\left(b(\psi^\dagger)^2 S_{H-i+j-1} + b^\dagger S_{H-i+j+1} + a(\psi^\dagger) S_{H-i+j}\right)$$

$$= (\psi^\dagger)^{i-j}\left(-|\psi|^2 S_{H-i+j-1} - S_{H-i+j+1} + (|\psi|^2 + 1) S_{H-i+j}\right)$$

$$= 0$$

**Case D**: Lastly lets consider the first column, i.e. $j = 0$ and $i > 0$. (The case of the last column follows as it is the complement and hence equal to 0.)

$$
\begin{aligned}
&\left[\mathbb{G}(\psi)[\mathbb{G}(\psi)]^{-1}\right]_{i,j} \\
&= \alpha \cdot (\psi^\dagger)^i S_\psi(H - i) + b \cdot (\psi^\dagger)^{i-1} S_\psi(H - i + 1) + \beta \psi^{H-i-1} S_\psi(i + 1) \\
&= 0.
\end{aligned}
$$

$\square$

## F  Proof of Strong Convexity(Lemma 4.2): Multi-dimensional

*Proof of Lemma 6.1.* Building on Section 6, we prove Lemma 6.1 for multi-dimensional systems. Using the fact that $E[w_i w_j^\top] = 0$ for different $i, j$ and $\mathbb{E}[w_i w_i^\top] = \Sigma$, it can be observed that for any $k_1, k_2$ and any $d_u \times d_u$ matrix $P$, we have that

$$
\mathbb{E}[J_{v_{\tilde{k}_1}}^\top P J_{v_{\tilde{k}_2}}] = \mathcal{T}_{k_1 - k_2} \otimes P \otimes \Sigma \tag{F.1}
$$

where $\mathcal{T}_m$ is defined as an $H \times H$ matrix with $[\mathcal{T}_m]_{ij} = 1$ if and only if $i - j = m$ and 0 otherwise. This in particular immediately gives us that for any matrix $P$,

$$
\mathbb{E}[J_y^\top P J_y] = \left( \sum_{k_1=1}^{H} \sum_{k_2=1}^{H} \mathcal{T}_{k_1-k_2} \otimes \left( \left( B^\top (A - B\mathbb{K})^\top \right)^{k_1-1} P (A - B\mathbb{K})^{k_2-1} B \right) \right) \otimes \Sigma
$$

$$
= \left( (I_H \otimes B^\top) \underbrace{\left( \sum_{k_1=1}^{H} \sum_{k_2=1}^{H} \mathcal{T}_{k_1-k_2} \otimes \left( ((A - B\mathbb{K})^\top)^{k_1-1} P (A - B\mathbb{K})^{k_2-1} \right) \right)}_{\triangleq \mathbb{G}_P} (I_H \otimes B) \right) \otimes \Sigma \tag{F.2}
$$

Furthermore consider the following calculation

$$
\mathbb{E}[J_{\tilde{v}_0}^\top \mathbb{K} J_y] = \left( \sum_{k=1}^{H} \mathcal{T}_{-k} \otimes \mathbb{K}(A - B\mathbb{K})^{k-1} B \right) \otimes \Sigma \tag{F.3}
$$

$$
= \left( (I_H \otimes \mathbb{K}) \underbrace{\left( \sum_{k=1}^{H} \mathcal{T}_{-k} \otimes (A - B\mathbb{K})^{k-1} \right)}_{\triangleq \mathbb{Y}} (I_H \otimes B) \right) \otimes \Sigma \tag{F.4}
$$

As before, we state the following bounds on the spectral properties of the matrices $\mathbb{G}$ and $\mathbb{Y}$ defined above.

**Lemma F.1.**

$$
\|\mathbb{Y}\| \le \| \sum_{k=1}^{H} \mathcal{T}_{-k}(A - B\mathbb{K})^{k-1} \| \le \gamma^{-1} \kappa^2 \tag{F.5}
$$

**Lemma F.2.** $\mathbb{G}_I$ *(where I represents the Identity matrix) is a symmetric positive definite matrix with*

$$
\mathbb{G}_I \succeq \frac{1}{4\kappa^4} \cdot I_{Hd_x}
$$

Consider the following calculations which follows by definitions.

$$
\begin{aligned}
\mathbb{E}[J_v^\top J_v] &= \mathbb{E}[J_y^\top \mathbb{K}^\top \mathbb{K} J_y] - \mathbb{E}[J_y^\top \mathbb{K}^\top J_{\tilde{v}_0}] - \mathbb{E}[J_{\tilde{v}_0}^\top \mathbb{K} J_y] + \mathbb{E}[J_{\tilde{v}_0}^\top J_{\tilde{v}_0}] \\
&= \underbrace{\left( (I_H \otimes B^\top) \mathbb{G}_{\mathbb{K}^\top \mathbb{K}} (I_H \otimes B) - \mathbb{Y}(I_H \otimes B) - (I_H \otimes B^\top) \mathbb{Y}^\top + I_{Hd_u} \right)}_{\triangleq \mathbb{F}} \otimes \Sigma
\end{aligned}
$$

Since we know that $\Sigma \succeq 0$ we immediately get that $\mathbb{F} \succeq 0$. Using the above calculations it is enough to show that the following matrix has lower bounded eigenvalues, i.e. for every vector $m$ of appropriate dimensions, we have that

$$m^\top \left( \mathbb{F} + (I_H \otimes B^\top) \mathbb{G}_I (I_H \otimes B) \right) m \geq \frac{\gamma^2 \|m\|^2}{36\kappa^{10}}$$

To prove the above we will consider two cases. The first case is when $\|(I_H \otimes B)m\| \geq \frac{\gamma \|m\|}{3\kappa^3}$. In this case note that

$$m^\top \left( \mathbb{F} + (I_H \otimes B^\top) \mathbb{G}_I (I_H \otimes B) \right) m \geq m^\top \left( (I_H \otimes B^\top) \mathbb{G}_I (I_H \otimes B) \right) m \geq \frac{\frac{1}{4\kappa^4} \gamma^2 \|m\|^2}{9\kappa^6}$$

In the second case (when $\|(I_H \otimes B)m\| \leq \frac{\gamma \|m\|}{3\kappa^3}$), we have that

$$
\begin{aligned}
&m^\top \left( \mathbb{F} + (I_H \otimes B^\top) \mathbb{G}_I (I_H \otimes B) \right) m \\
&\geq m^\top \left( I_{Hd_u} - (I_H \otimes \mathbb{K}) \mathbb{Y} (I_H \otimes B) - (I_H \otimes B^\top) \mathbb{Y}^\top (I_H \otimes \mathbb{K}^\top) \right) m \\
&\geq (1/3)\|m\|^2 \geq \frac{\gamma^2 \|m\|^2}{36\kappa^{10}}.
\end{aligned}
$$

$\square$

We now finish the proof with the proof of Lemmas F.1 and F.2.

*Proof of Lemma F.1.* Since $\mathbb{K}$ is $(\kappa, \gamma)$-diagonal strongly stable, we can diagonalize the matrix $A - B\mathbb{K}$ as $A - B\mathbb{K} = QLQ^{-1}$ with $\|Q\|, \|Q\|^{-1} \leq \kappa$. Therefore,

$$\mathbb{Y} = \left( \sum_{k=1}^H \mathcal{T}_{-k} \otimes QL^{k-1}Q^{-1} \right) = (I_H \otimes Q) \left( \sum_{k=1}^H \mathcal{T}_{-k} \otimes L^{k-1} \right) (I_H \otimes Q^{-1}).$$

Now consider the matrix $P$ for any complex number $\phi$ with $|\phi| < 1$.

$$P = \sum_{k=1}^H \mathcal{T}_{-k} \phi^{k-1}$$

We wish to bound $\|P\|$. To this end consider $PP^\top$ and consider the $\ell_1$ norm of any row. It can easily be seen that the $\ell_1$ norm of any row of $PP^\top$ is bounded by $\frac{1}{1-|\phi|} \cdot \frac{1}{1-|\phi|^2}$, and therefore

$$\|P\| = \sqrt{\|PP^\top\|} \leq \sqrt{\frac{1}{(1-|\phi|)(1-|\phi|^2)}}.$$

Using that $L$ is diagonal with entries bounded in magnitude by $1 - \gamma$, we get that $\|\mathbb{Y}\| \leq \gamma^{-1}\kappa^2$. $\square$

*Proof of Lemma F.2.* We need to consider the following matrix

$$\mathbb{G}_I = \sum_{k_1=1}^H \sum_{k_2=1}^H \mathcal{T}_{k_1-k_2} \otimes \left( \left( (A - B\mathbb{K})^\top \right)^{k_1-1} (A - B\mathbb{K})^{k_2-1} \right)$$

Since $\mathbb{K}$ is $(\kappa, \gamma)$-diagonal strongly stable, we can diagonalize the matrix $A - B\mathbb{K}$ as $A - B\mathbb{K} = QLQ^{-1}$ with $\|Q\|, \|Q\|^{-1} \leq \kappa$. Further since $A - B\mathbb{K}$ is a real valued matrix we have that $(A - B\mathbb{K})^\top = (Q^{-1})^\dagger L^\dagger Q^\dagger$. Therefore we have that

$$\mathbb{G}_I = \sum_{k_1=1}^H \sum_{k_2=1}^H \mathcal{T}_{k_1-k_2} \otimes \left( (Q^{-1})^\dagger \left( L^\dagger \right)^{k_1-1} Q^\dagger Q L^{k_2-1} Q^{-1} \right)$$

Further consider the following matrix $\hat{\mathbb{G}}$.

$$
\hat{\mathbb{G}} = \begin{bmatrix}
0 & 0 & . & . & I \\
0 & . & . & . & L \\
. & . & . & . & L^2 \\
. & 0 & . & . & . \\
0 & I & . & . & . \\
I & L & . & . & L^{H-1} \\
L & L^2 & . & . & 0 \\
L^2 & . & . & . & 0 \\
. & . & . & . & . \\
. & L^{H-1} & . & . & . \\
L^{H-1} & 0 & . & . & 0
\end{bmatrix}
$$

It can be seen that,

$$
\left( (I_{2H-1} \otimes Q) \hat{\mathbb{G}} (I_{2H-1} \otimes Q^{-1}) \right)^{\dagger} \left( (I_{2H-1} \otimes Q) \hat{\mathbb{G}} (I_{2H-1} \otimes Q^{-1}) \right) = \mathbb{G}_I. \tag{F.6}
$$

Furthermore note that since $\|Q\|, \|Q^{-1}\| \le \kappa$, therefore all singular values of $Q$ lie in the range $[\kappa^{-1}, \kappa]$. Therefore it follows that

$$
Q^{\dagger} Q \succeq \kappa^{-2} I \qquad (Q^{-1})^{\dagger} Q^{-1} \succeq \kappa^{-2} I \tag{F.7}
$$

Using (F.6),(F.7) it follows that

$$
\mathbb{G}_I \succeq \kappa^{-4} \cdot \left( \hat{\mathbb{G}} \right)^{\dagger} \left( \hat{\mathbb{G}} \right) \tag{F.8}
$$

Therefore we only need to show that $\left( \hat{\mathbb{G}} \right)^{\dagger} \left( \hat{\mathbb{G}} \right)$ has a lower bounded eigenvalue. To that end notice that since $L$ is a diagonal matrix with diagonal values whose magnitude is upper bounded by 1. Therefore, it sufficient to consider the case when $L$ is a scalar complex number with magnitude upper bounded by 1. To this end we can consider the following simplification of $\mathbb{G}_I$ defined for a complex number $\psi$ with $|\psi| < 1$ as defined earlier.

$$
\mathbb{G}(\psi) = \sum_{k_1=1}^{H} \sum_{k_2=1}^{H} \mathcal{T}_{k_1 - k_2} \left( \psi^{\dagger} \right)^{k_1 - 1} \psi^{k_2 - 1}
$$

Invoking Lemma 6.3 we immediately get that

$$
\mathbb{G}_I \succeq \kappa^{-4} \cdot \left( \hat{\mathbb{G}} \right)^{\dagger} \left( \hat{\mathbb{G}} \right) \succeq \frac{1}{4\kappa^4} \cdot I_{Hd_x}.
$$

$\square$