[Reviews · NeurIPS 2019]

Reviewer 1



This paper extends previous work for online control of observable linear systems under a strongly stable assumption and online strongly convex losses. The main contribution of this work is new analysis techniques that can exploit the strong convexity of the cost function, and as such allow first order methods to obtain faster rates. As in previous works, the main theoretical tool is to reduce the learning problem to an OCO with memory problem by considering a specific policy class with finite memory. Results are reused from past papers that argue the sufficiency of this policy to represent arbitrary linear policies (when the strong stability assumption holds). The main result is that the regret upper bound has significantly been improved from sqrt(T) to polylog(T), and the loss function can by strongly convex (with a know condition number). Given the recent activity on this problem, I think this paper presents an important contribution. The reduction to OCO with memory is a nice idea, and this paper makes another gets more mileage out of this idea. Is strong-stability necessary for your policy class to be close to optimal? Is there intuition there? Also, can the authors provide a description of the analytical innovations that were necessary to apply OGD (and specifically what was the barrier before)?

Reviewer 2



The paper concerns linear dynamical systems under the full knowledge of its dynamics, but with adversarially changing strongly convex costs. Two online/iterative algorithms are proposed which attain regret poly-logarithmic in time horizon T. On the one hand, this is somewhat expected as strongly convex losses lead to fast rates in online learning; on the other hand, the problem here is much more challenging as the past actions of the controller affect the future states of the system. The main trick is to parametrize the controller to depend linearly on the past H noise terms (called "disturbance-action policy" class); this goes almost without loss of generality as it has been shown in [4] that such parametrization allows to approximate any strongly-stable linear policy. The crucial fact shown here is that such parametrization retain strong convexity of the costs. Interestingly, this is due to the randomness and independence of the noise terms, so that they form a non-degenerate representation basis for controllers being their linear functions. Once strong convexity is established, the remaining part is to reduce to problem to online convex optimization with memory. This leads to two algorithms that attain poly-logarithmic regret: online gradient descent and online natural gradient. While the authors claim they are efficient, from what I understand they still need to compute the ideal cost function (Def. 3.4) and its gradient, which requires taking expectation of the cost over noise variables. This seems easily doable in some cases (e.g., Gaussian noise + quadratic costs), but might be quite complicated in some other cases. Up to my best knowledge, achieving logarithmic regret in this particular setup is a novel and original result, which should be of interest to (at least) theory part of machine learning community. I wish the authors included an instantiation of their algorithm for some nice setup, such as the case of quadratic cost function with Gaussian noise variables, as otherwise the algorithms look quite abstract. But this in no way diminishes their interesting theoretical result. This ends the positive part of the review. I found the paper very hard to follow. The only exception is the introduction, which is very well written and clear. However, once the paper goes into details of problem setting, it becomes dense in definitions and equations with very little accompanying explanation.A large part of it looks like a compressed version of [4], and I eventually ended up reading that paper as well, as otherwise I would not be able to grasp the motivation behind disturbance-state transfer matrix, ideal state and action, etc. I don't think this is how a good conference paper should look like. Furthermore, the proofs are also compressed to an extent that it sometimes requires quite some effort to check their validity (e.g., proofs of Theorem 4.1 and 4.2). To summarize, the paper is too dense to deliver its main idea clearly without the need to read past work. Moreover, the paper heavily relies on the result obtained in [4]. In particular, the disturbance-action policy representation with its approximation properties, the reduction to OCO with memory, analysis of the algorithms by means of ideal state, actions and costs are not novel, but rather directly taken from [4]. This somewhat diminishes the overall contribution. Having said that I still think the paper should be accepted if the authors revise the presentation of their results in the final version. One more remark: the inverse dependence on the bounds on the covariance of the noise sounds counter-intuitive. If anything, I would expect the bound improve when the noise is small. This is probably due to additional factor W^4 in front of the bound which controls the magnitude of the noise. I wonder a more intuitive factor would show up in the bound if the authors assumed, e.g., a (sub-)Gaussian noise. ---------------------------------------- After reading the authors' response ---------------------------------------- Thank you for posting a detailed rebuttal. All my questions were adequately answered. After reading the other reviews, it seems that I am the only one who complained so much about the presentation. Thus, I believe this might be partially due my smaller expertise in control of dynamical systems, and I increased my score by one to account for this. I still think that including a simple example of a dynamical system with an instantiation of your algorithm (such as the mentioned example of 1-d system) would be beneficial.

Reviewer 3



BRIEF SUMMARY ------------------------- In this work, the authors study the online control of known LDS under adversarially changing strongly convex cost functions. They extend the work of [1] as they deal with noisy dynamics, and generically strongly convex cost functions. However, the regret is still defined comparing with the best linear controller in hindsight. They show that the strong convexity is, as in the non-dynamical case, the key to obtain "fast-rate" i.e., logarithmic regret and they provide two algorithms (OGD and ONG) that achieve it. The main idea of the proof is to preserve the strong convexity following the approach initiated by [4] as opposed to the SDP relaxation of [9]. It consists in considering an over-parametrized policy, which linearly depends on the previous noise terms, up to lag $H$ that preserves the strong convexity and can approximate any stable linear policy in term of cost. With this Disturbance-Action Policy class, they connect to the Online Convex Optimization (OCO) with memory framework to design the algorithm and derive the regret analysis. They do so by considering "ideal" states, actions and costs, which discard the effect of previous states at a given lag. Such approximation however can be well controlled provided that the system is stable and that the lag $H$ is big enough. GENERAL REMARKS ------------------------------ The paper is well written and relatively easy to follow even though it uses a lot of material and notions from [4] and [5]. The result is novel and significantly improving over known results (from $\sqrt{T}$ to $\log(T)$. COMMENTS AND QUESTIONS ------------------------------------- - The result crucially relies on the preservation of strong convexity which was discarded in the SDP relaxation of [9]. It would be great to have a more detailed discussion on this matter, maybe recalling briefly the approach of [9], providing a comparison of approaches and sketching how a disturbance-action policy overcomes the removal of strong convexity. - The Asm on the cost functions are generic as it only requires strong convexity. However, the policy used for the comparator (in the regret definition) remains linear, which makes sense when the cost functions are quadratic (at least in the non-adversarial case). Is there any high-level arguments which could speak in favor of linear controller in the strongly convex case (as a 'good' policy) ? - I do not fully understand the need for Asm. 2.2 for the OGD regret bound. The LDS is known, and the Disturbance-Action Policy class allows a linear controller $K$ which, under controllability assumption can 'pre-stabilize' the system. As a result, the system is governed by $\tilde{A}$ which is a stable matrix. Could it be possible to replace Asm. 2.2. with a controllability assumption and to use a non-zero $K$ (stabilizing) in Thm. 4.1 ? If not, where does the proof break? - What is the controller $K$ used in ONG (it does not appear in the statement of Thm. 4.2.) ? I also think that the choice of $K$ should be given in the algorithm (either as an input, either in the two different instances). - It seems that there is a typo in Cor. 4.3. Should Asm 2.3 be replaced by 2.2 ? REFERENCES ------------------- [1] Yasin Abbasi-Yadkori, Peter Bartlett, and Varun Kanade. Tracking adversarial targets. [4] Naman Agarwal, Brian Bullins, Elad Hazan, Sham M Kakade, and Karan Singh. Online control with adversarial disturbances. [5] Oren Anava, Elad Hazan, and Shie Mannor. Online learning for adversaries with memory: price of past mistakes. [9] Alon Cohen, Avinatan Hasidim, Tomer Koren, Nevena Lazic, Yishay Mansour, and Kunal Talwar. Online linear quadratic control.

[Author Response · NeurIPS 2019]

We thank the expert reviewers for their high quality reviews and their appreciation of the results. We have answered a few (excellent) technical questions below, and hope that in light of this, the reviewers would consider raising their scores.

**0A: Novelty & Innovation:** Cohen et al. propose solving the problem on the state-action covariance as an online **linear** optimization routine, a class for which the best known upper and lower bounds scale as $\sqrt{T}$. We use alternative convex parameterization for the controller, suggested in Agarwal et al. However, this formulation employs a **highly over-parameterized** representation for the controller (as evident via parameter counting) rendering the program potentially not strongly convex in the parameters, even if the costs are strongly convex in $(x, u)$. This is the **main technical challenge**.

In this context, our results for OGD quantify the conditions under which the over-parameterization does not hinder strong convexity. Those for ONG exploit the specifics here to quantify more general conditions under which curvature is preserved in directions that matter for regret minimization. We'll add a detailed discussion to the paper upon revision.

**0B: More general conditions enabling logarithmic regret for OGD:** In actuality, the conditions that permit $\log T$ regret for OGD can be extended using the same techniques as presented in the paper with a more intricate analysis; given the reviewers' interest we will spell these out precisely in the final version. The result relaxes the assumption of stability of $A$ to a more general notion of controllability:

**Assumption 1.** [Diagonal Strong Stability] A controller $K$ is diagonally strongly stable if there exists a *complex* diagonal matrix $L$ and a complex matrix $Q$ such that $A - BK = QLQ^{-1}$ with $\|Q\|, \|Q^{-1}\| \leq \kappa$ and $\|L\| \leq 1 - \gamma$.

While the above is possibly more stringent than strong stability (Cohen et al), the fact that the above assumption is over complex diagonalization makes it quite general. The above extension comes about via a careful fine grained analysis of the Hessian aimed at bounding the effect (on strong convexity) of the time delayed influence of noise on controls introduced by including $Kx$ in the control.

**Reviewer #1** Please see **0A**.

**1A: On strong stability:** In stochastic systems, stable policies, of which strong stability is a quantification, ensure that the size of state ie. $\|x\|_2$ is bounded (independent of $T$). In this way, they permit operating with bounded instantaneous cost, which is a standard assumption in online learning.

**Reviewer #3** Please see **0A**.

**3A: Presentation & Writing:** We absolutely agree with the importance of readability for theoretical results. To ensure the paper can be read as a stand-alone, we thought to include a running example of 1-d system to illustrate the proof ideas. Perhaps the reviewer can comment on this presentation idea.

**3B: Computations concerning the ideal cost:** As noted for Gaussian noise with quadratic costs, the gradient and function value are analytically computable. Beyond this, we can get a close estimate by averaging samples. OCO algorithms are robust to $poly(T)^{-1}$ errors in gradient and function value.

**3C: Scaling with Noise:** Indeed, the regret scales as square of the noise (as it should since the cost is strongly convex), as long as the condition number of noise covariance is fixed.

**Reviewer #4** Please see **0A** and **0B**.

**4A: Linear policies beyond quadratic costs:** Beyond quadratic costs, as noted, the optimal policy might not be linear. But even the offline computation of such a policy poses computational difficulties. Therefore, we compare with and execute linear policies. It is a great future direction to find a good policy comparator class more generally.

**4B: On Assumption 2.2 for OGD:** Very insightful question. While it is true that the superimposition of a stable controller $K$ renders the effective dynamics $A' = A - BK$ stable, the cost associated with executing a disturbance-action policy is not the same on the original system $A$ (with superimposed $K$) and the system $A'$. This is because in the former, the cost (say $\|u_t\|^2$) associated with the action $u_t$ pays for the actions of $K$ in addition to the actions undertaken by the disturbance-action policy. This additional part to the cost makes the analysis particularly hard as it might reduce or eliminate the strong convexity in the disturbance-action parametrization.

However, as stated in **0B**, our technique is robust enough to tackle this issue, thereby extending generality of the results via a more fine-grained eigen characterization of the hessian.

**4C: Typos:** Indeed, Corollary 4.3 employs assumption 2.2. We agree that $K$ should be an input to the algorithm. Note that when working with Assumption 2.2 (e.g. OGD), $K = 0$ suffices, and when working with Definition 2.4 (i.e. ONG), any strongly stable $K$ suffices; the search for the later can posed as SDP feasibility as in Cohen et al.

[Meta-Review · NeurIPS 2019]

All reviewers highly appreciated the results and techniques in the paper, so it is definitely worthy of acceptance. The only complaint was that the paper is perhaps difficult to read, with some derivations being difficult to follow. The authors suggested to address this issue by illustrating the main ideas on a simple running example. I urge the authors to implement this change in the final version of the paper.